# Adult-onset temporal lobe epilepsy suspicious for autoimmune pathogenesis: Autoantibody prevalence and clinical correlates

**Julia C. Kuehn[1], Carolin Meschede[2], Christoph Helmstaedter[2,3], Rainer Surges[2,3], Randi von Wrede**🄲[2,3], **Elke Hattingen[4], Hartmut Vatter[5], Christian E. Elger[2,3], Susanne Schoch[1,2]ⵁ, Albert J. Becker[1]ⵁ, Julika Pitsch**🄲[1,2]ⵁ*

**1** Section for Translational Epilepsy Research, Dept. of Neuropathology, University Hospital Bonn, Bonn, Germany, **2** Dept. of Epileptology, University Hospital Bonn, Bonn, Germany, **3** Center for Rare Diseases Bonn (ZSEB), University Hospital Bonn, Bonn, Germany, **4** Dept. of Neuroradiology, University Clinic of Frankfurt, Frankfurt, Germany, **5** Clinic for Neurosurgery, University Hospital Bonn, Bonn, Germany

ⵁ These authors contributed equally to this work.

\* jpitsch@uni-bonn.de

**Data Availability Statement:** All relevant data are within the manuscript and its Supporting Information files.

## Abstract

Temporal lobe adult-onset seizures (TAOS) related to autoimmunity represent an increasingly recognized disease syndrome within the spectrum of epilepsies. In this context, certain autoantibodies (autoABs) were often associated with limbic encephalitis (LE). Here, we aimed to gain insights into (a) the distribution of 'neurological' autoABs (neuroABs, defined as autoABs targeting neuronal surface structures or 'onconeuronal' ABs or anti-glutamate acid decarboxylase 65 (GAD65) autoABs) in a large consecutive TAOS patient cohort, to characterize (b) clinical profiles of seropositive versus seronegative individuals and to find (c) potential evidence for other autoABs. Blood sera/cerebrospinal fluid (CSF) of TAOS patients (n = 800) and healthy donors (n = 27) were analyzed for neuroABs and screened for other autoABs by indirect immunofluorescence on hippocampal/cerebellar sections and immunoblots of whole brain and synaptosome lysates. Serological results were correlated with clinico-neuropsychological features. 13% of TAOS patients (n = 105) were neuroAB+, with anti-GAD65 and anti-N-methyl-D-aspartate receptors (NMDAR) as most frequent autoABs in this group. In our screening tests 25% of neuroAB- patients (n = 199) were positive (screening+), whereas all control samples were negative (n = 27). Intriguingly, key clinico-neuropsychological characteristics including magnetic resonance imaging (MRI) findings, epileptiform electroencephalographic (EEG) activity, and inflammatory cellular infiltrates in CSF were shared to a greater extent by neuroAB+ with neuroAB-/screening+ patients than with neuroAB-/screening- patients. Serological testing in a large consecutive TAOS patient series revealed seropositivity for anti-GAD65 autoABs as the most frequent neuroAB. Intriguingly, neuroAB+ individuals were virtually indistinguishable from neuroAB-/screening+ patients in several major clinical features. In contrast, neuroAB-/screening- TAOS patients differed in many parameters. These data support the potential presence of so far unrecognized autoABs in patients with TAOS.

**Funding:** This work was supported by Else Kröner-Fresenius-Stiftung (2016_A05 to JP, MD-stipend EKFS-Promotionskolleg 'NeuroImmunology' to JCK, AJB), the Deutsche Forschungsgemeinschaft (SFB 1089 to AJB, SS; FOR 2715 to AJB, SPP 1757 to SS) & the BONFOR program (AJB, SS, JP) and a Junior Researcher Group (JP) of the University of Bonn Medical Faculty.

**Competing interests:** The authors have declared that no competing interests exist.

## Introduction

Several neurological syndromes are linked to autoantibodies (autoABs) in serum and/or cerebrospinal fluid (CSF) targeting different proteins [1, 2]. These include the disease spectrum of limbic encephalitis (LE), the definition of which encompasses temporal lobe seizures, subacute early adult-onset memory impairment and/or affective disturbances [3–6]. Clinical findings in LE are associated with characteristic magnetic resonance imaging (MRI) changes involving amygdaloid and hippocampal structures as well as a range of neuropathological alterations comprising lymphocytic inflammation of limbic structures and hippocampal sclerosis (HS) [7].

LE variants relate to the presence of specific autoABs in serum and/or CSF [8] and can develop as paraneoplastic [9] or non-paraneoplastic conditions [10, 11]. LE-patients are stratified according to the presence of non-paraneoplastic autoABs directed against neuronal surface structures involving N-methyl-D-aspartate receptors (NMDARs), voltage-gated potassium channel complex (VGKC) components including Leucine-rich glioma inactivated 1 (LGI1) and Contactin associated protein 2 (CASPR2), A-amino-3-hydroxy-5-methyl-4-isoxazolepropionic acid receptor (AMPARs) and C-aminobutyric acid receptor A/B (GABA$_{A/B}$Rs) [11–16]. Onco-neural autoABs include anti-amphiphysin, -CV2 and -PNMA2 (Ma2/Ta; paraneoplastic antigen Ma2) autoABs [17]. Anti-glutamic acid decarboxylase 65 (GAD65) autoABs occur in a generally non-paraneoplastic condition and target intracellular protein structures [18].

The criteria of 'limbic syndrome' have been recently defined in a strict manner [19]. Compared to patient cohorts from general neurological or neuro-oncological institutions studied for autoAB-related encephalitis, tertiary epileptology centers are subject to a different patient series selection bias. Epileptologists are mainly confronted with adult patients presenting difficult to explain new onset temporal lobe epilepsies as leading symptom. Those patients share many, but not all features of what currently is declared as essential for the diagnosis of LE. Here, we report on a large consecutive series of patients newly referred to a large Epilepsy Center over more than three years suffering from temporal lobe adult-onset seizures (TAOS) with clinical findings suggestive of an autoimmune origin. Compared to previous studies on ABs in highly selected epilepsy patient cohorts [20–23], here, we have assessed for the first time the clinical results in a patient group, in which the presence of autoABs is suspected but has not been identified yet, and compared this group to patients positive for 'neuroABs'.

## Materials and methods

### Patients, serum and CSF samples

Biofluids of 800 patients with TAOS (youngest patient included was 18 years of age), presented in the Department of Epileptology, University Hospital Bonn, a tertiary epilepsy centre (visited by ~1000 inpatients and ~5000 outpatients per year), between 11/2013 and 12/2016, were included in this study. We only included patients in this study, which fulfilled the following criteria: (a) temporal lobe seizures of unknown etiology with onset in adulthood and (b) at least one other feature predicting autoimmune caused epilepsy including impaired episodic memory, substantial affective disturbances, characteristic MRI and/or CSF changes.

With respect to relevant MRI changes, a large fraction of patients showed otherwise unexplained medio-temporal Fluid-attenuate inversion recovery (FLAIR)/T2 signal increase and/or increased volume in mesial structures of one or both temporal lobes [19]. 966 serum and 434 CSF samples were analyzed. For TAOS patients with repetitive tests at our center, the biological sample with the highest autoAB titer has been considered in this study. Serological testing comprised established diagnostic procedures as well as additional tests for other autoAB than previously known neuroABs. Initially, sera/CSF-samples were tested for neuroABs using cell-based

assays as well as immunoblots described elsewhere in detail [19]. Furthermore, we developed an integrated screening paradigm that combined immunoblotting and indirect immunofluorescence tests (IIFT) in order to scrutinize for the presence of further autoABs. Serum samples from 27 healthy donors (male: n = 13; female: n = 14; mean age: 34 years ± 10.6 SD) were used as controls. All control samples were handled and tested parallel to the patient biosamples. Sera from four healthy controls that showed reactivity to HEp2 cell assays indicating the presence of anti-nuclear antibodies (ANA) were excluded from further analysis as these autoantibodies are apparently not relevant to LE since in these individuals any neurological symptoms are lacking. Positive and negative controls for commercial assays were provided by the manufacturer.

### Ethics statement

All procedures were conducted in accordance with the Declaration of Helsinki. This study was approved by the Ethical Commission of University Hospital Bonn (222/16). Informed written consent was obtained from every patient. We strictly adhered to the legal provisions governing the admissibility of the handling of personal data.

### Collection of clinical data

Relevant clinical information was extracted from medical reports. Clinical data of three groups was collected retrospectively. Prior to further analysis, all personal data was anonymized. We analyzed patients (a) seropositive for neuroAB (neuroAB$^+$), (b) seronegative for well-known neuroABs at the time point of data collection but with positive results in either one screening test pointing to potentially novel autoABs (neuroAB$^-$/screening$^+$) and (c) entirely seronegative for autoAB (neuroAB$^-$/screening$^-$). With respect to MRI alterations, patients were stratified according to changes suspicious for LE including mesio-temporal volume and/or FLAIR/T2 signal increase [19, 24], hippocampal sclerosis (HS) [25] and "other lesions" comprising cavernoma, cortical atrophy, arachnoid cyst or vascular lesions. All MRI images were reviewed by an experienced neuro-radiologist. Patients with electroencephalographic (EEG) analyses revealing a defined focus at least at one time point were classified as 'EEG focus'-positive. Lateralization of this focus was included in the analyses. The seizure type was classified by an experienced epileptologist according to the recent classification of the International League Against Epilepsy (ILAE) [26]. Inflammatory CSF parameters comprised elevated protein level (>400 mg/l), pleocytosis (>5 cells/μl) as well as oligoclonal bands (OCB) or intrathecal Immunglobulin G (IgG) synthesis (according to the 'Reiber scheme'). Memory impairment was assessed by standardized neuropsychological tests of the verbal and figural memory described elsewhere in detail [27] conducted by psychologists of the Department of Epileptology. Values one standard deviation below average were classified as memory impairment.

Detailed information concerning comorbidities including autoimmune disorders, malignancies, psychosis, motoric symptoms as well as depression were retrospectively collected from clinical reports psychosis comprised symptoms of delusion, significant impairment of perception and/or 'ego'-disturbances according to the 'ADMP'-system. In order to define depression, the International Statistical Classification of Diseases and Related Health Problems 10[th] Revision (ICD10) F32 criteria were applied as standard. Motor symptoms unrelated to seizures included stiff-person-syndrome, neuromyotonia, ataxia and spasticity.

### Semiquantitative immunoblotting and cell-based indirect immunofluorescence tests for neuroABs

Using commercial diagnostic kits for, all analyzed serum and CSF samples were screened for neuroABs known to be associated with LE or other neuroimmunological syndromes. In detail,

we tested for 'onconeuronal' autoABs against Amphiphysin, CV2, PNMA2 (Ma2/Ta; paraneoplastic antigen Ma2), Ri, Yo, Hu, Recoverin, SOX1, Titin, Zic4 (Zinc finger protein Zic4), GAD65 (Glutamate acid decarboxylase 65) and Tr (DNER) using semiquantitative immunoblots according to the manufacturer's guidelines (EUROLINE PNS 12, Euroimmun, DL 1111-1601-7 G) coated with recombinant antigen or antigen fragments with serum diluted 1:100 and CSF 1:1 (sensitivity: 89–100%%; specificity: 99%, according to the manufacturer's specifications). Complementarily, indirect immunofluorescence was performed using on HEK293T-cells (Human Embryonic Kidney 293) overexpressing of the individual antigens on the cell-surface (IIFT: 'Autoimmune-Enzephalitis-Mosaik1', Euroimmun, FA 1120-1005-1; GAD65-IIFT, Euroimmun, FA 1022-1005-50) for anti-NMDAR, anti-CASPR2, anti-LGI1, anti-GABA$_A$R, anti-GABA$_B$R, anti-AMPAR and anti-GAD65 autoABs (dilution: serum 1:10; CSF 1:1) following the manufacturer's protocol (sensitivity: 82–100%; specificity: 98,3–100%, according to the manufacturer's specifications). The evaluator was blinded to the particular clinical phenotype. A positive result in one of the commercial assays was sufficient to classify the patient seropositive for neuroAB. NeuroABs with targets previously described as relevant in other neuroimmunological syndromes including myasthenia (anti-Titin) or retinopathy (anti-Recoverin) [28, 29] were excluded from our analyses.

## Screening for potentially novel autoABs in sera/CSF

In parallel to the analyses outlined above, we have established a screening approach for the presence of novel autoABs using immunoblots. To this end, lysates of rat and mouse brain extracts, of human hippocampal tissue from pharmacoresistant temporal lobe epilepsy (TLE) patients undergoing epilepsy surgery for seizure relief [30] and of murine crude synaptosomes (**see S1 File**) were prepared, separated by electrophoresis and blotted. After blocking [2% bovine serum albumin (BSA), 2% fetal calf serum (FCS), 0.2% cold water fish gelatin in phosphate-buffered saline (PBS)], blots were incubated with patient's serum (1:500) or CSF (1:100) in a total volume of 7 ml overnight, washed with PBS with Tween 20 (PBST), incubated with goat anti-human IRDye 800CW (Odyssey, 926–32232) for 45 minutes and imaged with the Odyssey Imaging System (LI-COR) after another washing step.

For corresponding IIFT screenings, we used a custom-made biochip-based-assay (IIFT: 'Neurologie-Mosaik28', Euroimmun, FA 111-1005-28) including rat and simiiform slices of cerebellum and hippocampus in order to screen for binding patterns of potential autoABs to neuronal tissue (dilution: serum 1:10, CSF 1:1) according to the manufacturer protocol. The custom-made assays were only available after the first 295 biosamples of the present series. These samples have already been analyzed by a highly similar approach relying on permeabilized human hippocampal, rat forebrain and cerebellar 10 μm sections (**see S1 File**). All sera suspicious for autoAB binding in one of the screening assays were subsequently tested on HEp2 (Human epithelioma 2)-cell-assays (IIFT: HEp2; Euroimmun; FA 1520–1005) for reactivity with non-neuronal antigens (dilution 1:100) according to the manufacturer's instructions. Images of the IIFT assays were taken with an epifluorescence microscope (Observer.A1; Zeiss). All immunoblots and IIFT-assays were analyzed by an experienced examiner (AJB).

## Statistical analysis

Statistical comparisons with respect to individual clinical features (categorical variables) between neuroAB$^+$, neuroAB$^-$/screening$^+$, neuroAB$^-$/screening$^-$ and healthy controls were performed using Chi-square test or Fisher's exact test (GraphPad Prism, Version 6.07, GraphPad Software, Inc.). P-values were adjusted by Bonferroni correction (Microsoft Excel 2010). Values were considered significantly at p<0.05. Power-analysis was carried out using the g-power

software. All clinical parameters to be examined were determined prior to data acquisition and statistical testing.

## Results

### Distribution of neuroABs in TAOS patients

We analyzed 1400 biological samples of 800 TAOS patients (male: n = 402; female: n = 398; mean age: 44.5 years ± 16 SD) collected during a time period of three years (11/2013-12/2016). In 13% of all patients at least one neuroAB was detected (neuroAB[+]). 23.8% of these patients had neuroABs targeting cell-surface associated antigens including NMDAR with the highest prevalence (16.2%) as well as LGI (4.8%) or CASPR2 (2.9%; Table 1, Fig 1A). NeuroABs against intracellular GAD65 were detected in 46.7% of all neuroAB[+] patients representing the highest prevalence within the spectrum of all neuroABs (Table 1). 55.1% of these anti-GAD65 autoAB positive patients had high titers (≥1:100) representing 25.7% of all neuroAB[+] patients (Fig 1A). 29.5% of neuroAB[+] patients were positive for neuroABs with reactivity to other intracellular antigens including amphiphysin, CV2, Ma2/Ta and SOX1. We detected five anti-Yo and three anti-Zic4 autoAB positive patients in the present series; all of them presented focal impaired awareness seizures (up to five per day) and only one patient had a mild ataxic gait in addition to seizures. Five of them had imaging findings showing the involvement of mesio-temporal structures. The majority of sera (31 out of 32) from patients with neuroABs had been detected in CSF was also tested positive. We detected only one exception, i.e. an anti-Ri autoAB patient was only CSF-positive. None of the control sera showed evidence for the presence of neuroABs.

**Hippocampus and cerebellum-based IIFT screening in neuroAB- TAOS patients.** 62% of all patients were seronegative for neuroABs and did not show any reactivity in screening assays, neither IIFT nor immunoblots (neuroAB[-]/screening[-]). IIFTs revealed binding patterns in 18.5% (122 out of 661) of neuroAB[-] patients' biomaterial (Fig 1B and 1C). This observation suggests that autoABs distinct from neuroABs might be present in CSF or serum samples of

**Table 1. Presence and distribution of neuroABs in 800 TAOS patients.**

| Target | Patients | Percen-tage[a] | Age[b] limbic (mean ± SD) | Sex (m:f) | CSF[c] | Titer Serum[d] | Titer CSF[d] |
|---|---|---|---|---|---|---|---|
| LGI1 | 5 | 4.76% | 60.8 ± 13.55 | 4:1 | 1/3 | 1:10–1:320 | 1:10 |
| CASPR | 3 | 2.86% | 51 ± 5.72 | 1:2 | 1/1 | 1:10–1:1000 | 1:1 |
| NMDAR | 17 | 16.19% | 41.24 ± 13.5 | 4:13 | 2/11 | 1:10–1:320 | 1:10 |
| Amphi-physin | 3 | 2.86% | 38.33 ± 23.44 | 0:3 | 0 | + | - |
| CV2 | 2 | 1.90% | 30 ± 9.89 | 2:0 | 0 | + | - |
| Ma2/Ta | 12 | 11.43% | 45.83 ± 15.78 | 4:8 | 5/8 | + | (+) |
| Ri | 2 | 1.89% | 47.5 ± 2.12 | 1:1 | 1/1 | + | + |
| SOX1 | 4 | 3.81% | 47 ± 21.18 | 1:3 | 1/2 | + | + |
| Yo | 5 | 4.76% | 52.8 ± 19.79 | 2:3 | 0 | + | - |
| Zic4 | 3 | 2.86% | 61 ± 13.75 | 2:1 | 1/1 | ++ | +++ |
| GAD 65 | 49 | 46.67% | 44.06 ± 16.12 | 18:31 | 20/30 | 1:10–1:3200 | 1:1–1:100 |

SD = standard deviation; m = male; f = female; Metric variables are given as mean values; non-metric data are presented as medians.

[a] Percentage of neuroAB[+] patients (n = 105).

[b] Age (years) at diagnosis

[c] number of positively tested CSF samples of all tested CSF samples which were available in parallel to sera

[d] range of titers, lowest titer considered positive was 1:10 for serum and 1:1 for CSF, semiquantitative analyses of immunoblots range from (+) = marginal positive, + = positive, ++ = strongly positive to +++ = extremely positive.

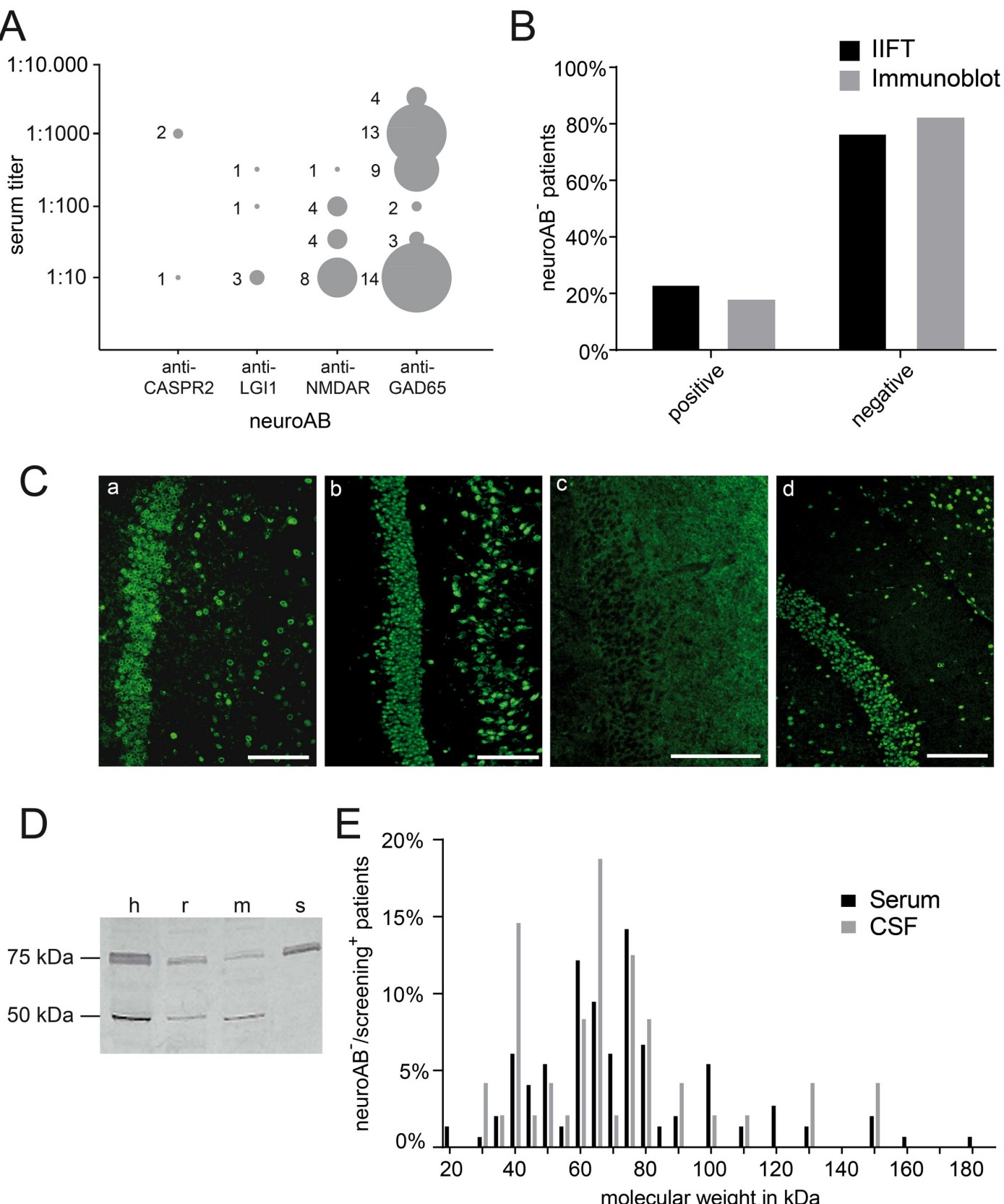

**Fig 1. Representative indirect immunofluorescence Test (IIFT) and immunoblotting results.** (**A**) Distribution of autoAB titers in neuroAB+ patients. Circle sizes are proportional to the number of patients with the respective titres (1:10, 1:32, 1:100, 1:320, 1:1000 or 1:3200). (**B**) ~20% of neuroAB- patients are positive either with immunoblot and/or IIFT screening indicating the presence for so far unknown autoABs. (**C**) Binding patterns of serum samples of neuroAB- patients on representative simiiform hippocampal sections: (**a**) neuronal cytoplasmatic (**b**) general neuronal (**c**) neuropil and (**d**) neuronal nuclear binding pattern (all scale bars: 100 μm). (**D**) Representative immunoblot of serum from a neuroAB- patient. Reactivity with lysate of brain from different species is illustrated (h = human, r = rat, m = mouse, s = murine synaptosomes). Two bands (75 kDa and 50 kDa) occur in human, rat and mouse brain lysates. The synaptic fraction shows a band at 75 kDa but not at 50 kDa identifying the putative autoAB-target within the 75 kDa band as component of the synaptic compartment. (**E**) Distribution of the molecular weight of bands observed in immunoblots of neuroAB-/screening+ patients' sera and CSF samples. Note the clustering of bands in the 40–80 kDa range (serum: n = 145, CSF: n = 47).

these TAOS patients. In 12 patients only CSF samples but not the corresponding sera were IIFT positive. This finding may be in line with an intrathecal synthesis of immunoglobulins. Many patients had a similar IIFT pattern in paired serum and CSF samples. The IIFT pattern was distinguished in neuropil (n = 36), general neuronal (n = 44), and neuronal nuclear versus cytoplasmatic (n = 21, each) binding patterns of serum/CSF samples (Fig 1C a-d). Most of the patients' biomaterial showed specific reactivity in limbic structures (50.4% of samples were IIFT-positive *only* in hippocampal tissue); positivity in both, cerebellum *and* hippocampus, was less frequent (24.6% of all IIFT-positive samples). None of the control sera were positive in the IIFT screening test.

**Whole brain and synaptosome based immunoblot screening in neuroAB- TAOS patients.** Complementarily, we analyzed 876 sera and 310 CSF samples of 662 neuroAB- patients by immunoblotting. In biological samples of 100 neuroAB- TAOS patients at least one additional band was detected compared to control serum samples (Fig 1B). In 88% of these blots the patient specific additional band was found in the mouse synaptosome lysate, suggesting the presence of autoABs targeting synaptic molecules (rightmost lane, Fig 1D). More than 53% of the corresponding CSF samples paralleled the positive findings in sera. For 73% of sera with reactivity in the immunoblot screening the signal was observed in the human hippocampal lysate (leftmost lane, Fig 1D).

The molecular weight of the majority of proteins revealed by incubation with patient sera was in the ranges of 40–80 kDa (Fig 1E). Two or more patient specific bands in immunoblots were present in sera of 10% of neuroAB- TAOS patients positive in immunoblot screening. In contrast, in CSF samples multiple bands occurred only in 2%. Most bands detected after incubation with CSF exhibited a similar size spectrum (Fig 1E). 28.6% (n = 199) neuroAB- TAOS patients were positive in one screening test, either IIFT or immunoblot (neuroAB-/screening+ patients). 12% of these 199 patients revealed positive results in *both* screening assays. Immunoblotting as well as IIFT screenings serologically distinguished controls from neuroAB+ patients (see S1 Fig; healthy donors n = 27, neuroAB+ patients n = 105, Fisher´s exact test: immunoblot: ***p<0.0001; IIFT: ***p<0.0001). Additionally, none of the control samples was positive in the immunoblot screening test.

Sera from patients with positive test results in one of the above mentioned screening assays were examined by immunofluorescence of non-neuronal cells (HEp2-cells) to identify antibodies against ubiquitously expressed proteins. In those patients, the seroprevalence of antibodies binding HEp2-cells was 44.6%. The prevalence of a positive assay did not differ among the groups of neuroAB+ and neuroAB-/screening+ patients (44.4% vs. 44.7%, neuroAB+: n = 72, neuroAB-/screening+: n = 172; chi square test group comparison: p = 0.766). 42.3% of sera from patients with positive results in the immunoblot screening and 50.5% of patients with positive results in the IIFT showed HEp2-cell reactivity.

**Correlation of clinico-pathological and serological findings in TAOS patients.** All patients of the present series had a disease onset in adult life defined by temporal lobe seizures and at least one of the following features including (a) disturbance of memory, (b) affective

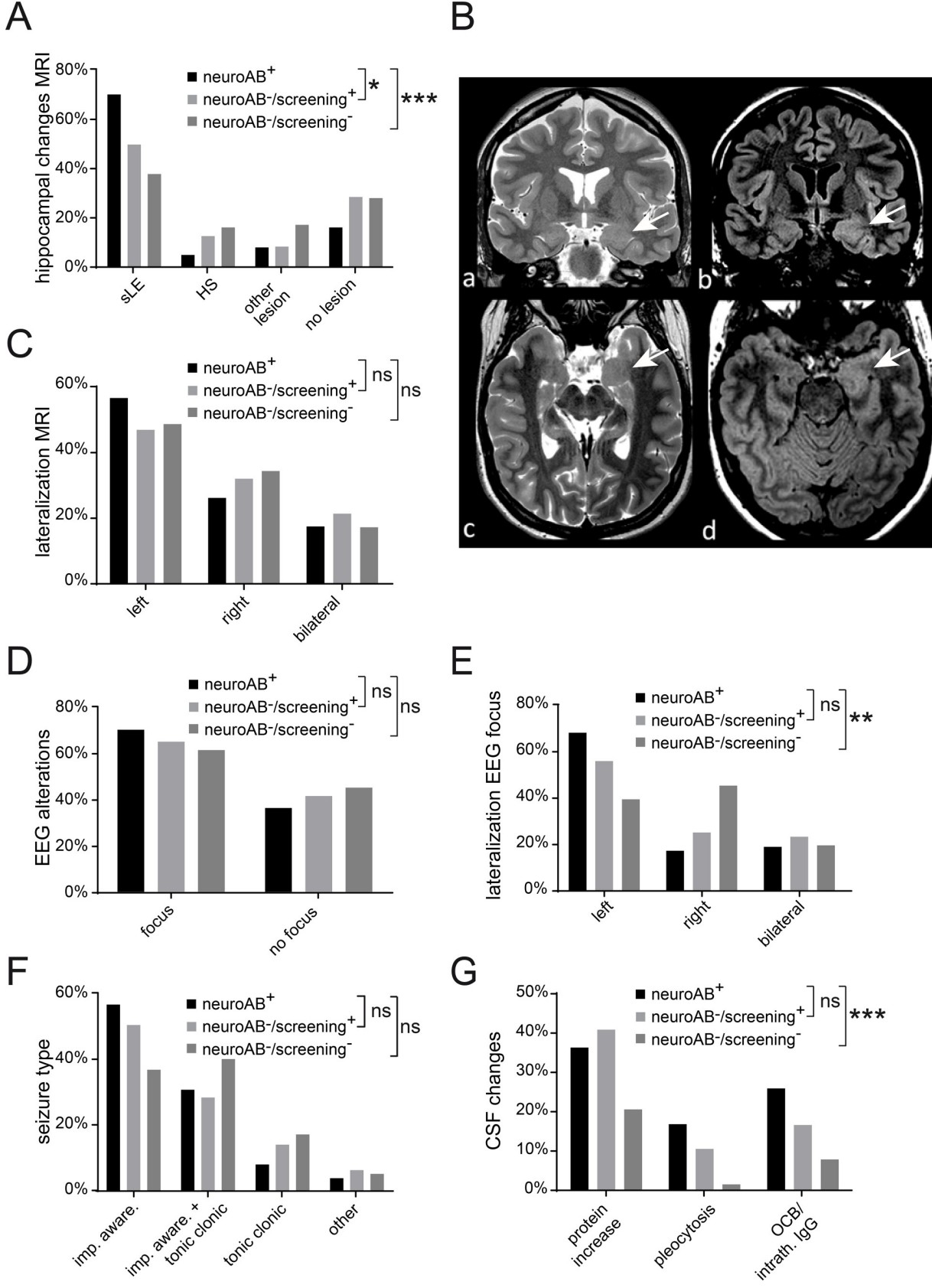

**Fig 2. Imaging, semiological and serological parameters of TAOS patients stratified according to autoAB-test status.** (**A**) NeuroAB[+] patients have substantially more frequent LE suspicious MRI changes (sLE) comprising signs of edema in amygdala and hippocampus than neuroAB[+] and neuroAB[-]/screening[+] patients (Chi-square-test group comparison: [*]p<0.025; [***]p<0.0005; neuroAB[+] n = 98, neuroAB[-]/screening[+] n = 94, neuroAB[-]/screening[-] n = 92). (**B**) Representative MRI in a patient with limbic encephalitis and anti-GAD65 autoAB. Coronar T2-weigthed MRI (**a**) and FLAIR (**b**) shows swelling and signal increase of the left hippocampus (see white arrows) compared to the contralateral hippocampus. The temporal angulated axial T2-weigth image (**c**) and the axial FLAIR (**d**, ac-pc angulation) confirms the increase in volume and size of the left compared to the right hippocampus (see white arrows). (**C**) The lateralization of MRI changes does not differ between neuroAB[+] and neuroAB[-]/ screening[+] (Chi-square-test group comparison: p = 0.589) or neuroAB[+] and neuroAB[-]/screening[-] (Chi-square-test group comparison: p = 0.79; neuroAB[+] n = 69, neuroAB[-]/screening[+] n = 47, neuroAB[-] n = 35). (**D**) In most of the patients an EEG focus is detectable with no significant differences between the neuroAB[+] and neuroAB[-]/screening[+] (Chi-square-test group comparison: p = 0.544) or neuroAB[+] and neuroAB[-]/ screening[-] (Chi-square-test group comparison: p = 0.288; neuroAB[+] n = 91, neuroAB[-]/screening[+] n = 95, neuroAB[-]/screening[-] n = 92). (**E**) A difference concerning affected hemisphere is observed in neuroAB[+] patients compared to neuroAB[-]/screening[-] patients (Chi-square-test group comparison: [**]p<0.005) but not between neuroAB[+] patients and neuroAB[-]/screening[+] (Chi-square-test group comparison: p = 0.424; neuroAB[+] n = 60, neuroAB[-]/screening[+] n = 60, neuroAB[-]/screening[-] n = 53). (**F**) Most neuroAB[+] patients have focal impaired awareness seizures in some patients also focal to bilateral tonic clonic seizures occur (Chi-square-test group comparison: neuroAB[+] compared to neuroAB[-]/screening[+] p = 0.477; neuroAB[+] compared to neuroAB[-]/screening[-] p = 0.039; neuroAB[+] n = 97, neuroAB[-]/screening[+] n = 91, neuroAB[-]/screening[-] n = 92). (**G**) In comparison to neuroAB[+] patients typical CSF changes [elevated protein level (protein increase), pleocytosis, oligoclonal bands and intrathecal IgG synthesis (intrath. IgG)] are present in a similar frequency in neuroAB[-]/screening[+] patients (Chi-square-test group comparison: p = 0.866). In contrast neuroAB[-]/screening[-] patients have significantly less inflammatory CSF changes (Chi-square-test group comparison: [***]p<0.0005; neuroAB[+] n = 77, neuroAB[-]/screening[+] n = 66, neuroAB[-]/screening[-] n = 63). sLE–limbic encephalitis suspicious, HS–hippocampal sclerosis, imp. aware–impaired awareness, OCB–oligoclonal bands.

disturbances, or (c) characteristic temporal findings on MRI or in CSF. The time from symptom-onset to the detection of a specific neuroAB was 6.3 ± 8 SD years. This parameter strongly differed among the different neuroAB[+] subgroups. Whereas all anti-LGI1 or anti-CASPR2 autoAB positive patients were diagnosed within the first year, patients with anti-GAD65 auto-ABs experienced substantially longer delays (6.1 ± 5 SD years). The time period between the onset of symptoms and the testing for neuroABs was 7.9 ± 10 SD years in neuroAB[-]/screening[+] patients and 9.7 ± 10 SD years in neuroAB[-]/screening[-] patients. In this time period temporal lobe epilepsy with no obvious cause was diagnosed in the majority of these patients (neuroAB[-]/screening[+]: 58%; neuroAB[-]/screening[-]: 68%).

In approximately 70% of neuroAB[+] TAOS patients, characteristic MRI alterations were present including increased volume and T2-signal intensity of the mesio-temporal structures (representative MRI scan in Fig 2B), but the presence differed significantly between the groups (Chi-square-test group comparison: [*]p<0.025; [***]p<0.0005). Whereas limbic encephalitis suspicious (sLE) MRI findings comprising signs of edema in amygdala and hippocampus were substantially less frequent in neuroAB[-]/screening[+] and neuroAB[-]/screening[-] patients, HS was more frequent in the two latter groups than in neuroAB[+] patients (Fig 2A). No differences were found with respect to MRI-lateralization (Fig 2C) as well as the presence of an EEG focus (Fig 2D). However, the lateralization analysis of EEG alterations revealed a significant difference between neuroAB[+] and neuroAB[-]/screening[-] patients (Fig 2E, Chi-square-test group comparison: [**]p<0.005). Semiologically, most neuroAB[+] patients suffered from focal seizures. With respect to seizure types no differences were found between the groups (Fig 2F). In CSF pleocytosis, oligoclonal bands (OCB), and intrathecal IgG synthesis was substantially more frequent in neuroAB[+] and neuroAB[-]/screening[+] patients compared to neuroAB[-] individuals (Fig 2G, Chi-square-test group comparison: [***]p<0.0005).

Most patients of the present series (70.7%) had memory impairment based on standardized neuropsychological testing (Fig 3A). In neuropsychological tests of figural as well as verbal memory, no lateralization predominance of cognitive impairment was present. Nevertheless, these tests were overall below the average of healthy individuals in all groups of patients. A subset of neuroAB[+] patients (34.3%) suffered from neuropsychiatric impairment including psychosis (Fig 3B), depression (Fig 3C), or motoric disorders (Fig 3D). NeuroAB[-]/screening[-] patients were significantly less affected from motor impairment than neuroAB[+] individuals

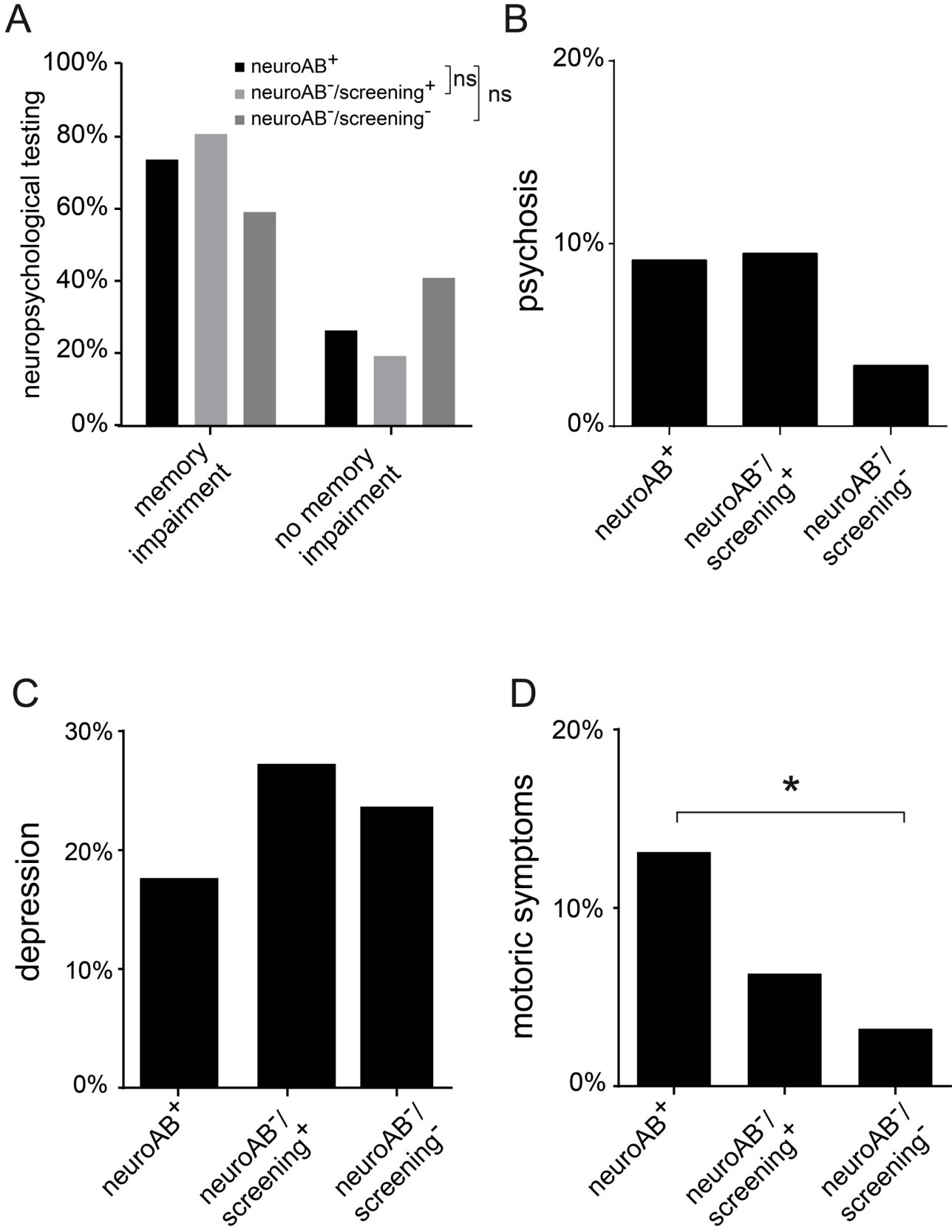

**Fig 3. Neurological and psychiatric co-morbidities in TAOS patients.** (**A**) Memory disturbance, i.e. neuropsychological test results below average, is a common feature throughout the patients stratified according to the serological status with no significant differences between the groups (Chi-square-test group comparison: neuroAB$^+$ compared to neuroAB$^-$/screening$^+$ p = 0.284; neuroAB$^+$ compared to neuroAB$^-$/screening$^-$ p = 0.044 neuroAB$^+$ n = 95, neuroAB$^-$/screening$^+$ n = 78, neuroAB$^-$/screening$^-$ n = 93) (**B, C**) Remarkably, psychosis and depression are not substantially different in frequency between the three groups of patients (Chi-square-test group comparison: psychosis: neuroAB$^+$ compared to neuroAB$^-$/screening$^+$ p = 1; neuroAB$^+$ compared to neuroAB$^-$/screening$^-$ p = 0.136/ depression: neuroAB$^+$ compared to neuroAB$^-$/screening$^+$ p = 0.461; neuroAB$^+$ compared to neuroAB$^-$/screening$^-$ p = 0.209; neuroAB$^+$ n = 99, neuroAB$^-$/screening$^+$ n = 95, neuroAB$^-$/screening$^-$ n = 93). (**D**) In contrast, motoric symptoms are substantially more abundant in neuroAB$^+$ than in neuroAB$^-$/screening$^-$ patients (Chi-square-test group comparison: *p<0.025). No significant differences in the presence of these symptoms are discovered between the neuroAB$^+$ and neuroAB$^-$/screening$^+$ group (Chi-square-test group comparison: p = 0.147; neuroAB$^+$ n = 99, neuroAB$^-$/screening$^+$ n = 95, neuroAB$^-$/screening$^-$ n = 93).

(Chi-square-test group comparison: *p<0.025), whereas neuroAB$^-$/screening$^+$ patients were similar to the neuroAB$^+$ patients (Fig 3D).

With respect to systemic disorders in the present series of TAOS patients, we focused on the presence of general autoimmune syndromes as well as neoplastic disorders. Intriguingly, systemic autoimmunity was more abundant than neoplastic disorders (Fig 4A) with a significant abundance for the neuroAB$^+$ group (Fisher´s exact test: *p<0.016). The frequency of tumors was similar in patients with onconeuronal autoABs (13.3%; n = 4) and patients with autoABs against neuronal surface proteins (12%; n = 3; Fig 4A). Malignant tumors of different entities were observed in all analyzed groups (Table 2). We found an association with additional extracerebral autoimmune-mediated syndromes including autoimmune thyroiditis, chronic inflammatory bowel diseases and collagenosis/chronic polyarthritis (Fig 4B).

## Discussion

In this study we investigated the presence of established neuroABs and screened for the potential incidence of novel autoABs in patients with TAOS of suspected autoimmune pathogenesis but no presence of known neuroABs. So far, studies with a focus related to our present approach have put emphasis on the prevalence of established 'neurological' autoABs related to different, utmost important clinical aspects including new or established epilepsy. Whereas only patients fulfilling a battery of clinico-serological criteria were included in former series [6, 19, 22, 31, 32], the common clinical denominator of our present study is given by adult onset seizures that represent a vigorous clinical challenge for epileptologists. Nevertheless, there are forms of TLE which do not need to be tested for autoABs. Those include patients in which TLE can be clearly linked to by causes not related to autoimmune mechanisms including tumors or cortical dysplasias in the temporal lobe.

Our present study goes beyond serological testing for 'neurological' ABs by analyzing neuroAB negative patients but screening positive for autoABs putatively targeted against not yet characterized protein structures. The use of additional screening tests (IIFT / immunoblots) may be regarded supportive and complementary to specific autoAB tests in individuals with clinical characteristics of possible autoimmune encephalitis seropositive in screening assays but negative for specific neuronal or onconeuronal autoABs with the perspective to define new forms of autoAB related (limbic) encephalitis [19]. Recently, this strategy has led to the observation of a novel autoAB targeting the intracellular dendritic spine scaffolding protein Drebrin in so far neuroAB$^-$/screening$^+$ patients [33].

Individual autoABs have been analyzed in depth for sensitivity with respect to the particular type of biofluid, i.e. serum or CSF. In particular, for anti-NMDAR autoABs it has been demonstrated that there is higher 'clinical sensitivity' in CSF than in serum. Generally, antibody titers in CSF and serum were reported as higher in patients with poor outcome or teratoma than in patients with good outcome or no tumor. Intriguingly, titer change in CSF more closely reflected relapses than one in sera [34].

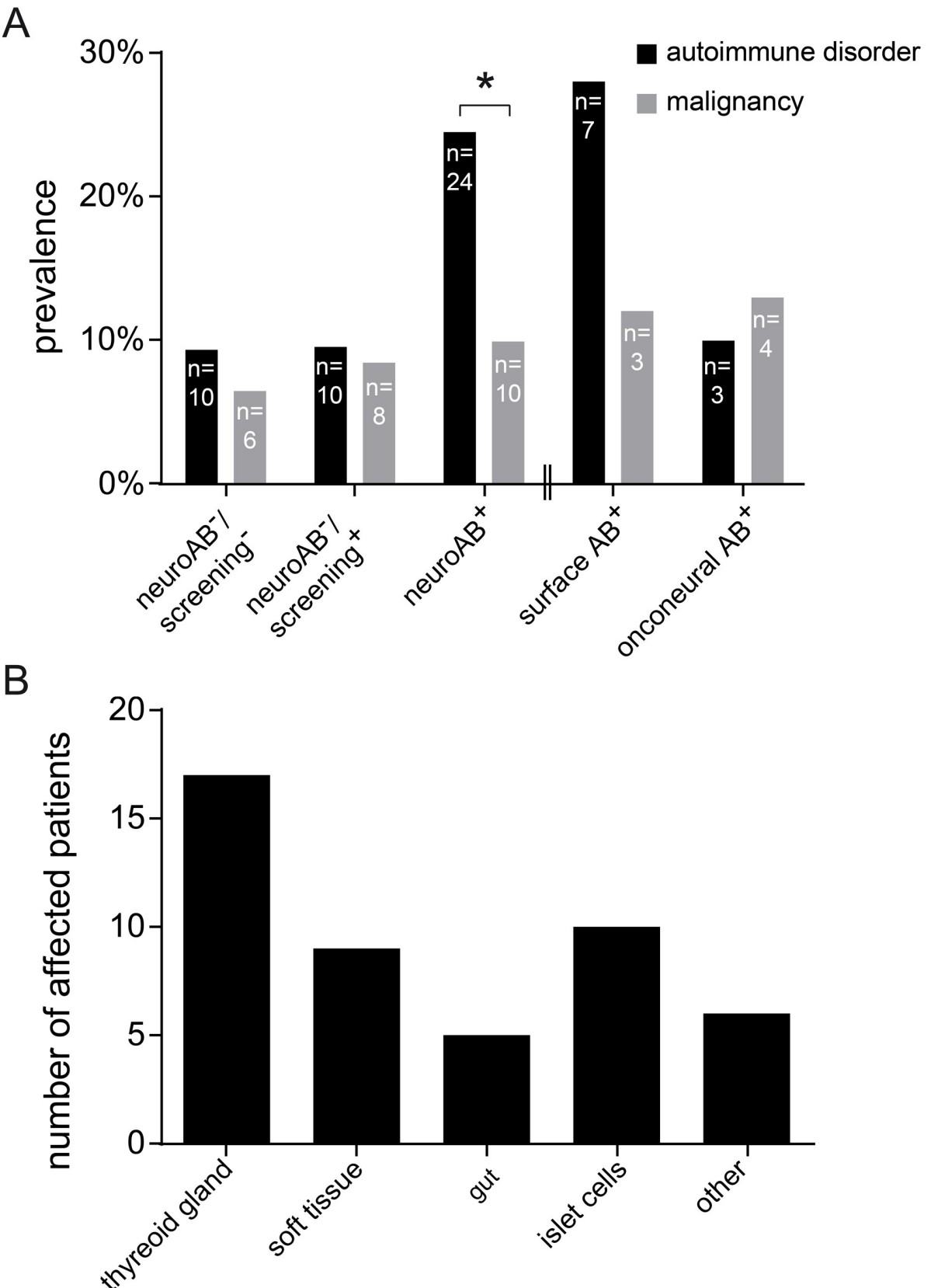

**Fig 4. Co-morbidities of TAOS patients.** (**A**) Systemic autoimmune syndromes are more abundant than neoplasia in all groups. The neuroAB[+] group has significant more autoimmune syndromes compared to neoplasia of different entities (Fisher´s exact test: *p<0.016; neuroAB[+] n = 101). This difference in abundance is not significant in the neuroAB[-]/screening[-] or neuroAB[-]/screening[+] group (Fisher´s exact test: neuroAB[-]/screening[+] patients: p = 0.805, n = 95; neuroAB[-]/screening[-] patients: p = 0.434, n = 93). No significant difference is present comparing co-morbidities of seropositive patients with autoABs targeting neuronal surface structures to patients with onconeural autoABs (Fisher´s exact test: p = 0.393; surface AB[+] n = 25, onconeural AB[+] n = 30). (**B**) Systemic autoimmunity comprises thyreoditis as most abundant disorder followed by chronic inflammatory bowel disease, collagenosis/chronic polyarthritis and diabetes mellitus type 1 as well as individual autoimmune disorders grouped under 'other' (overall n = 47 affected patients).

In the context of previous studies, two large epilepsy cohorts with either established or newly diagnosed untreated epilepsy were compared regarding the prevalence of key neurological autoABs in adult patients [20]. 11% of these patients had autoABs and such targeting VGKC complex proteins were most abundant (5%). As the data of Brenner and colleagues [20] as well as our present data were collected from patients referred to a tertiary center, it cannot be ruled out that the influence of certain selection biases influences the distribution of individual neuroABs in different cohorts. Nevertheless, the previously described significantly higher prevalence of anti-GAD65 autoABs in TLE in comparison to other seizure syndromes [35] is well in line with the higher prevalence of these autoABs in our study focusing solely on TAOS patients. Moreover, Dubey et al. reported on a higher prevalence for autoABs against VGKC complex molecular structures in a series of 112 epilepsy patients with recent onset or unknown etiology [36], which may be due to differences in patient cohorts.

In limbic encephalitis, the autoAB spectrum comprises 'onconeural' ABs including amphiphysin (anti-AMPH), BMP binding endothelial regulator (anti-BMPER; anti-CV2), paraneoplastic Ma antigen 2 (anti-Ma2; anti-PNMA2) as well as further anti- glutamic acid decarboxylase 65 (GAD65)9, all of them targeted against intracellular protein structures. GAD65 as molecule with intracellular localization as well as the neuropathological observation of an admixture of neurodegenerative changes with invasion of T-lymphocytes in the biopsy tissue of patients [18] may suggest analogies of the present disease pattern with paraneoplastic neurological syndromes. Within our current patient cohort, anti-Ma2/Ta autoAB positive patients had the highest prevalence within the 'onconeuronal' autoAB-positive group of patients. Here, specific onconeuronal antigens including Ta/M proteins are expressed constitutively in neurons and ectopically in the associated tumors. In brain lesions, neurodegeneration is accompanied by perivascular infiltrates constituted of CD4[+] T-cells and B-cells, whereas CD8[+] T-cells predominate within the CNS parenchyma [37, 38]. Purely onco-

**Table 2. Distribution of malignancies in TAOS patients.**

| Group | Patients[a] | Malignancy |
|---|---|---|
| Anti-LGI1-autoAB | 1/5 | colorectal carcinoma |
| Anti-NMDAR-autoAB | 2/17 | colorectal carcinoma, medulloblastoma, basal cell carcinoma |
| Anti-Ma2/Ta-autoAB | 2/12 | ALL, acini cell carcinoma |
| Anti-Yo-autoAB | 1/5 | breast cancer |
| Anti-Zic4-autoAB | 1/3 | AML, prostate carcinoma |
| Anti-GAD65-autoAB | 3/49 | breast cancer, cervical carcinoma, multiple myeloma |
| neuroAB-/screening+ | 9/95 | breast cancer, multiple myeloma, ALL, bladder cancer, oesophageal cancer, neuroendocrine tumour, oligoastrocytoma, astrocytoma |
| neuroAB-/screening- | 6/93 | breast cancer, prostate carcinoma, Hodgkin lymphoma, renal cell carcinoma, glioblastoma multiforme |

[a] number of patients affected by malignancies in relation to total number of patients in this autoAB group. ALL = acute lymphoblastic leukemia; AML = acute myeloid leukemia.

neuronal antibody-dependent pathogenetically relevant mechanisms are doubtful in view of the intracellular localization of most onconeuronal antigens and a consistent failure to replicate the CNS pathology of paraneoplastic neurological syndromes by antibody transfer in an animal model [39–41]. More recently, pathological and immunological investigations implicate the T-cell component of the lesions as the primary pathogenic principle in these disorders [37, 42–44]. In an exemplary fashion, it was shown that by adoptive transfer, T-cells specific for the autologous onco-neuronal antigen Pnma1 elicited a strong neuroimmunological reaction in contrast to minor pathology by anti-Pnma1 antibodies induced by protein immunization. These data indicate that in paraneoplastic neurological syndromes pathogenetically critical T-lymphocytes can target identical molecules such as the corresponding onconeuronal antibody does [45]. Significant effects of anti-GAD65 antibodies on neuronal function or synaptic transmission have not been found under experimental conditions [46, 47]. In contrast to our data, a recent study on patients with paraneoplastic LE revealed higher anti-Hu autoAB prevalence [9]. Although autoABs targeting Yo and Zic4 are mainly known for their association with paraneoplastic ataxia, individual cases of patients presenting features of LE have been described [48, 49]. This applies also to all anti-Yo as well as anti-Zic4 autoAB positive patients in this study presenting clinically typical LE findings.

Intriguingly, we found systemic autoimmune disorders in parallel to TAOS more frequent than tumors with a significant increase in the neuroAB⁺ group. In none of the patients, systemic lupus erythematosus was diagnosed, which in fact represents a differential diagnosis of autoimmune-mediated epilepsy. Moreover, antinuclear antibodies (ANAs) including anti-double-stranded deoxyribonucleic acid (anti-dsDNA) or anti-Smith (anti-SM) were not detected in any of the patients' biomaterial. Patients having collagenosis as comorbidity have mixed connective tissue disease, scleroderma, and Sjögren's syndrome. The general predisposition of developing autoimmune disorders in some individuals apparently includes autoimmune seizure-related clinical manifestations. This aspect may affirm the previously described correlation between epilepsy and autoimmune diseases in a population-based study [50].

We further grouped patients that were neuroAB⁻ but positive in at least one of our screening assays as neuroAB⁻/screening⁺ and compared neuroAB⁺ to those as well as to neuroAB⁻/screening⁻ patients. We are fully aware of the fact that this stratification has limitations and represents a simplified approach since patients with pathogenetically different neuroABs are grouped. Despite these constraints, several clinical parameters including signs of inflammation in CSF, MRI findings and additionally lateralization of the EEG focus support a difference between neuroAB⁺ and neuroAB⁻/screening⁻ patients. neuroAB⁻/screening⁺ patients were only different from neuroAB⁺ when referring to hippocampal MRI characteristics. This difference may be due to the fact, that neuroAB⁺ individuals are diagnosed with LE rather early in the disease process.

In a sophisticated approach, Dubey et al. have established an "APE" score that reports on symptoms and clinical signs predictive of autoantibody positivity [36]. However, our present study has a fundamentally different focus by in depth comparisons of different patient groups according to the described battery of specific neuroAB tests as well as autoAB screenings. The International League Against Epilepsy 'Autoimmunity and Inflammation Taskforce' has recently suggested to differentiate between 'acute symptomatic seizures secondary to autoimmune encephalitis' and 'autoimmune-associated epilepsy' [51]. Clinically, this conceptual distinction may be also helpful to learn about differences of the clinical course of neuroAB⁻/screening⁺ patients. Clearly, the results of autoAB testing have to be carefully interpreted in the context of clinical symptoms in individual patients.

The present data point to the importance of considering autoimmune mechanisms in patients with the leading symptom of adult onset temporal seizures may encourage further research on this topic and the development of tailored treatments.

## Supporting information

**S1 Fig. Screening for autoABs in TOAS patients and healthy donors.** Healthy donors had significantly less positive (**A**) immunoblot and (**B**) IIFT results in comparison to the TAOS patient cohort (TAOS patients n = 765, healthy donors n = 27, Chi-square-test group comparison: *p<0.05).
(TIF)

**S1 File. Supplemental methods.**
(DOCX)

**S2 File.**
(DOCX)

**S3 File.**
(PDF)

**S1 Raw image.**
(TIF)

## Acknowledgments

We thank Indra Prusseit for excellent technical assistance.

## Author Contributions

**Conceptualization:** Christoph Helmstaedter, Rainer Surges, Randi von Wrede, Christian E. Elger, Susanne Schoch, Albert J. Becker, Julika Pitsch.

**Formal analysis:** Julia C. Kuehn, Carolin Meschede, Christoph Helmstaedter, Elke Hattingen, Julika Pitsch.

**Funding acquisition:** Susanne Schoch, Albert J. Becker, Julika Pitsch.

**Investigation:** Julia C. Kuehn, Carolin Meschede, Christoph Helmstaedter, Rainer Surges, Randi von Wrede, Elke Hattingen, Hartmut Vatter, Christian E. Elger, Susanne Schoch, Julika Pitsch.

**Methodology:** Julia C. Kuehn, Albert J. Becker, Julika Pitsch.

**Project administration:** Albert J. Becker, Julika Pitsch.

**Resources:** Carolin Meschede, Rainer Surges, Randi von Wrede, Elke Hattingen, Hartmut Vatter, Christian E. Elger, Susanne Schoch, Albert J. Becker, Julika Pitsch.

**Supervision:** Albert J. Becker, Julika Pitsch.

**Visualization:** Julia C. Kuehn, Julika Pitsch.

**Writing – original draft:** Julia C. Kuehn, Susanne Schoch, Albert J. Becker, Julika Pitsch.

**Writing – review & editing:** Julia C. Kuehn, Rainer Surges, Randi von Wrede, Elke Hattingen, Susanne Schoch, Albert J. Becker, Julika Pitsch.

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
