## [Decision Letter · Decision Letter 0]

28 Aug 2020

PONE-D-20-21286

Adult-onset temporal lobe epilepsy suspicious for autoimmune pathogenesis: autoantibody prevalence and clinical correlates

PLOS ONE

Dear Dr. Pitsch,

Thank you for submitting your manuscript to PLOS ONE. After careful consideration, we feel that it has merit but does not fully meet PLOS ONE’s publication criteria as it currently stands. Therefore, we invite you to submit a revised version of the manuscript that addresses the points raised during the review process.

We look forward to receiving your revised manuscript.

Kind regards,

Giuseppe Biagini, MD

Academic Editor

PLOS ONE

Journal Requirements:

'All procedures were conducted in accordance with the Declaration of Helsinki and the Ethical Commission of University Hospital Bonn (222/16). Informed written consent was obtained from every patient.'

a. Please amend your current ethics statement to confirm that your named institutional review board or ethics committee specifically approved this study.

5. Please include captions for your Supporting Information files at the end of your manuscript, and update any in-text citations to match accordingly. Please see our Supporting Information guidelines for more information: http://journals.plos.org/plosone/s/supporting-information

6. Your ethics statement must appear in the Methods section of your manuscript. If your ethics statement is written in any section besides the Methods, please move it to the Methods section and delete it from any other section. Please also ensure that your ethics statement is included in your manuscript, as the ethics section of your online submission will not be published alongside your manuscript.

Reviewers' comments:

Reviewer's Responses to Questions

**Comments to the Author**

1. Is the manuscript technically sound, and do the data support the conclusions?

Reviewer #1: Yes

Reviewer #2: Yes

2. Has the statistical analysis been performed appropriately and rigorously? 

Reviewer #1: I Don't Know

Reviewer #2: Yes

3. Have the authors made all data underlying the findings in their manuscript fully available?

Reviewer #1: Yes

Reviewer #2: Yes

4. Is the manuscript presented in an intelligible fashion and written in standard English?

Reviewer #1: Yes

Reviewer #2: Yes

5. Review Comments to the Author

Reviewer #1: This is an interesting study on autoantibody prevalence at a tertiary epilepsy center. I have some comments.

Materials and methods:

The author needs to comment more on their selection of patients. Bonn seems to have a population of 320 000. Temporal lobe epilepsy is not more than perhaps on fifth of all epilepsies, so collection of 800 patients is really remarkable. The authors need to provide more detail on the size of the territory center, the number of adult epilepsy patients seen_/reviewed in total and perhaps also on the regional health care organization. Which patients are referred?

Related to the above, more information is needed on how patients were selected. Were they identified in an electronic database? Identified by treating phycisians? How robust is this procedure?

Perhaps I missed it, but I did not see a description of the ethical permission. This needs to be included, and also a statement on handling of personal data.

Regarding antibody tests: specify which commercial tests were used for which antibodies.

Was the evaluated of the antibody tests blinded to the clinical phenotype? If so, please state that in the methods.

Results

I don’’t think the difference between onconeroal Abs and surface Abs with regard to tumours (13.3% vs 12%) should be mentioned as a difference. It is a very similar level.

I'm a bit hesitant about the number of statistical comparisons. What about mass significance? Were all these comparisons specified pre-anaysis? This should perhaps be mentioned in the materials and methods.

Figure 2.

I think the ILAE term is not “unaware”, but "impaired awareness".

Discussion

The authors should elaborate more on what their results mean for clinicians:

Should testing be done more often or less often?

Should all TLE cases be screened?

Is serum testing sufficient or css required?

Overall, the study was very interesting to read. It adds considerable information to the field because of the number of patients.

Reviewer #2: Kuehn et all present an important paper on the diagnosis of autoantibody mediated Adult onset temporal lobe epilepsy and suggest methodology for further analysis of antibody negative patients. In these they find a significant proportion who have unidentified antibodies present. The manuscript is generally well written but has a few grammatical errors that could be picked up by a native English speaker (e.g. page 4 line 94-95, page 10 line 222).

As a methodological issue, I was surprised that patients with an ANA were excluded. While I certainly understand that this finding does make IIF difficult, it would have been interesting to know if there were associated with neuronal antibody specificity in the additional assays, which would not necessarily be affected by the ANA (mainly the Western blot). Additionally, there was no comment as to whether these patients would fit the criteria for CNS lupus. Which may have epilepsy as a presenting feature. This could easily be dealt with in the discussion. Also, in the methods there was sensitivity and specificity given for their assays and is was not clear where they were derived from. It was not clear whether the finding of positive standard onconeural antibodies was dependent on the finding of IIF in addition to Line blot positivity or whether a positive line blot alone was sufficient to call the antibody positive.

The finding of a high prevalence of anti-GAD antibodies was indeed interesting and this with the finding of increased family history of autoimmunity raises the issue as to whether the presence of an antibody is the primary cause of the TLE or whether the initial TLE has led to increased exposure of neuronal antigens to the peripheral immune system and loss of tolerance leading to autoantibody presence. Of course, this is a difficult question to answer; however, it may be worthy of discussion.

6. PLOS authors have the option to publish the peer review history of their article (what does this mean?). If published, this will include your full peer review and any attached files.

Reviewer #1: No

Reviewer #2: **Yes: **Dr David A Brown

---

## [Author Response · Author response to Decision Letter 0]

6 Oct 2020

Responses to the individual referees’ comments on the manuscript entitled " Adult-onset temporal lobe epilepsy suspicious for autoimmune pathogenesis: autoantibody prevalence and clinical correlates” (Manuscript ID: PONE-D-20-21286)

Referee 1.

We thank the referee for commending our work as ‘an interesting study on autoantibody prevalence at a tertiary epilepsy center’. The individual points of the reviewer were addressed in detail as follows:

Point 1: The author needs to comment more on their selection of patients. Bonn seems to have a population of 320 000. Temporal lobe epilepsy is not more than perhaps on fifth of all epilepsies, so collection of 800 patients is really remarkable. The authors need to provide more detail on the size of the territory center, the number of adult epilepsy patients seen/reviewed in total and perhaps also on the regional health care organization. Which patients are referred?

Response: We thank the reviewer for these questions. The Department of Epileptology of the Bonn University Hospital belongs to one of the tertiary epilepsy centre in Germany, in which a large number of national and international patients are diagnosed and treated. Our centre is situated in North Rhine Westfalia, which has one of the highest population densities in Germany. Here, a multitude of patients consult our specialized Epileptologists, consisting of 1000 inpatients and 5000 outpatients per year. Our specialized centre is mainly contacted by resident neurologists for specific epilepsy-related questions and topics including the differential diagnostics in general, syndrome diagnostics and/or drug-resistant epilepsy. We have now included a more detailed description on our patient cohort in the revised manuscript (p. 4, ll. 91-92).

Point 2: Related to the above, more information is needed on how patients were selected. Were they identified in an electronic database? Identified by treating physicians? How robust is this procedure?

Response: We included patients in this study which fulfill the following criteria (a) temporal lobe seizures of unknown etiology with onset in adulthood and (b) at least one other feature predicting autoimmune caused epilepsy including impaired episodic memory, substantial affective disturbances, characteristic MRI and/or CSF changes. Adult onset temporal lobe epilepsy (TLE) was diagnosed by examined senior Epileptologists by the use of: diagnostic examination, physical and psychological anamnesis, ECG, MRI, neuropsychological testing and blood analysis as well as CSF tests in a subset of patients. This combination of outstanding diagnostic approaches, intense case discussion in boards and high-frequent follow-up visits was validated (p. 4, ll. 93-96).

Point 3: Perhaps I missed it, but I did not see a description of the ethical permission. This needs to be included, and also a statement on handling of personal data.

Response: We moved the ethical statement into its own section within the Methods chapter as suggested by the Editor. We also added a statement on handling personal data in our manuscript (p. 5, ll. 115-119).

Point 4: Regarding antibody tests: specify which commercial tests were used for which antibodies. Was the evaluated of the antibody tests blinded to the clinical phenotype? If so, please state that in the methods.

Response: Commercial kits were used for diagnostic procedures including Amphiphysin, CV2, PNMA2 (Ma2/Ta; paraneoplastic antigen Ma2), Ri, Yo, Hu, and GAD65 (Glutamate acid decarboxylase 65) applying semiquantitative immunoblots according to the manufacturer’s guidelines (EUROLINE PNS 12, Euroimmun, DL 1111-1601-7 G) coated with recombinant antigen or antigen fragments with serum diluted 1:100 and CSF 1:1. Complementary, indirect immunofluorescence was used relying on HEK293-cells (Human Embryonic Kidney 293) with overexpression of the individual antigens on the cell-surface (IIFT: ‘Autoimmune-Enzephalitis-Mosaik1’, Euroimmun, FA 1120-1005-1; GAD65-IIFT, Euroimmun, FA 1022-1005-50) for NMDAR-, CASPR-, LGI1-, GABAA-, GABAB-, AMPAR- and GAD65-autoABs (dilution: serum 1:10; CSF 1:1) following the manufacturer’s protocol. The evaluator was blinded to the particular clinical phenotype. We included this additional information in the revised manuscript (p. 6, l.151 – p. 7. l. 166).

Point 5: Results: I don’t think the difference between onconeuronal Abs and surface Abs with regard to tumours (13.3% vs 12%) should be mentioned as a difference. It is a very similar level.

Response: We agree on the Reviewer’s point and have revised the corresponding paragraph (p. 16, ll. 385-387). 

Point 6: I' m a bit hesitant about the number of statistical comparisons. What about mass significance?

Response: We are aware of the problems caused by multiple testing. We therefore corrected for multiple testing by using the well-established Bonferroni-Method (p. 8, l. 198), a valid approach for Chi-square test (Wright, 1992; McDonald, 2014). 

Point 7: Were all these comparisons specified pre-anaysis? This should perhaps be mentioned in the Materials and methods.

Response: All clinical parameters to be assessed and analyzed have been determined prior to data acquisition and statistical testing. As suggested by the reviewer, we added a statement in the Materials and method section in order to address this comment (p. 9, ll. 200-201). 

Point 8: Figure 2: I think the ILAE term is not “unaware”, but "impaired awareness".

Response: We thank for this remark and changed the terminology in the manuscript (p. 9, l. 216).

Point 9: Discussion: The authors should elaborate more on what their results mean for clinicians: Should testing be done more often or less often?

Response: The referee addresses an important issue. In depth auto-antibody analysis appears reasonable in the case of possible autoimmune encephalitis (Graus et al., 2016). Thereby, clearly the results of auto-antibody testing have to be interpreted in the context of other clinical findings in individual patients. Those include subacute onset of working memory deficits, impaired mental status or psychiatric symptoms, novel focal alterations with emphasis of limbic structures, seizures that are unexplained by pre-described seizures syndromes, pleocytosis of CSF as well as MRI features proposing limbic encephalitis. The use of screening tests (IIFT / immunoblots) may be regarded supportive and/or complementary to testing for specific antibodies. Data of screening tests may confirm results of specific antibody testing e.g. in the case of anti-NMDAR autoantibodies, for which IIFT results of hippocampal sections are very characteristic. Furthermore, screening tests can be extremely helpful in defining new forms of auto-antibody related (limbic) encephalitis in patients seropositive in screening assays but negative for specific neuronal or onconeuronal antibodies as we have recently shown for anti-Drebrin antibodies in patients with new onset epilepsy in adult age and further clinical signs of limbic encephalitis (Pitsch et al., 2020). These considerations are now included in a condensed form in the revised manuscript (p. 18, ll. 425-429; p. 22, ll. 510-511).

Point 10: Should all TLE cases be screened?

Response: The underlying etiopathogenesis of TLE can be rather diverse (Blumcke et al., 2017). There are forms of TLE which should not prompt analysis for auto-antibodies. Those include patients in which TLE can be clearly explained by causes not related to autoimmune mechanisms including tumors or cortical dysplasias in the temporal lobe. As outlined before, particularly characteristics categorizing individuals as patients with possible autoimmune encephalitis should be carefully considered in the context of analyses for the presence of autoantibodies (Graus et al., 2016). These considerations are now included in a condensed form in the revised manuscript (p. 18, ll. 420-422).

Point 11: Is serum testing sufficient or csf required?

Response: The referee addresses another intriguing issue that in our view cannot be entirely resolved at this point. Individual autoantibodies have been analyzed in depth in this context. This holds particularly true for anti-NMDAR autoantibodies. Dalmau and colleagues have demonstrated that there is higher sensitivity of NMDA receptor antibody testing in CSF than in serum. Generally, antibody titers in CSF and serum were reported as higher in patients with poor outcome or teratoma than in patients with good outcome or no tumor. Intriguingly, the titer change in CSF more closely reflected relapses than was it was the case in sera (Gresa-Arribas et al., 2014). These arguments have been included in the revised manuscript (p. 19, ll. 432-437).

Referee 2.

We thank this referee for the constructive suggestions and for the comment that this is “an important paper on the diagnosis of autoantibody mediated Adult onset temporal lobe epilepsy...”.

Point 1: The manuscript is generally well written but has a few grammatical errors that could be picked up by a native English speaker (e.g. page 4 line 94-95, page 10 line 222).

Response: We corrected the mentioned phrases. Additionally, the entire manuscript was carefully revised by a native English speaker.

Point 2: As a methodological issue, I was surprised that patients with an ANA were excluded. While I certainly understand that this finding does make IIF difficult, it would have been interesting to know if there were associated with neuronal antibody specificity in the additional assays, which would not necessarily be affected by the ANA (mainly the Western blot). 

Response: None of the patients with ANA were excluded from the study. Only sera from 4 healthy controls that showed reactivity to HEp2 cell assays indicating the presence of anti-nuclear antibodies (ANA) as these autoantibodies are apparently not relevant to LE since in these individuals any neurological symptoms are lacking. 

All patients with positive test results in the additional assays were tested for antibodies targeted to non-neuronal cells (HEp2-cells). In 44.6% of these patient’s biofluids reactivity was observed. NeuroAB+ patients had similar frequencies of positive test results (44.4%) compared to neuroAB-/screening+ patients (44.7%). Patients with positive results in Western blot-Assay were also positive in Hep2-cell assays in 42.3% of cases. Regarding patients with positive result in IIFT, 50.5% of these patients had ANA. We included this additional information in the revised manuscript (p. 12, l. 285 – p. 13, l. 292).

Point 3: Additionally, there was no comment as to whether these patients would fit the criteria for CNS lupus. Which may have epilepsy as a presenting feature. This could easily be dealt with in the discussion.

Response: None of the patients included in the study has been diagnosed for systemic lupus erythematosus. Moreover, antinuclear antibodies (ANAs) including anti-double-stranded deoxyribonucleic acid (anti-dsDNA) or anti-Smith (anti-SM) were not detected in any of the patient’s biomaterial. Patients having collagenosis as comorbidity have mixed connective tissue disease, scleroderma, and Sjögren’s syndrome. We included an informative paragraph on this aspect in the discussion (p. 17, ll. 389-392; p. 21, ll. 482-487).

Point 4: Also, in the methods there was sensitivity and specificity given for their assays and is was not clear where they were derived from. It was not clear whether the finding of positive standard onconeural antibodies was dependent on the finding of IIF in addition to Line blot positivity or whether a positive line blot alone was sufficient to call the antibody positive.

Response: Sensitivity and specificity data were derived from the manufacturer’s specifications for the commercial antibody tests. According to the manufacturer 100 healthy blood donors were examined in order to determine the specificity. Sensitivity was determined by using 95 pre-characterized sera of patients with paraneoplastic neurological syndrome. 

A positive result in one of the two tests was sufficient to classify a patient seropositive for neuroAB. As the sensitivity of the tests is high, we considered one positive test result sufficient. We clarified our specifications in the material and method section (p. 7, ll. 157/158; 164).

Point 5: The finding of a high prevalence of anti-GAD antibodies was indeed interesting and this with the finding of increased family history of autoimmunity raises the issue as to whether the presence of an antibody is the primary cause of the TLE or whether the initial TLE has led to increased exposure of neuronal antigens to the peripheral immune system and loss of tolerance leading to autoantibody presence. Of course, this is a difficult question to answer; however, it may be worthy of discussion.

Response: In limbic encephalitis, the autoantibody spectrum comprises ‘onconeural’ ABs including amphiphysin (anti-AMPH), BMP binding endothelial regulator (anti-BMPER; anti-CV2), paraneoplastic Ma antigen 2 (anti-Ma2; anti-PNMA2) as well as further anti- glutamic acid decarboxylase 65 (GAD65)9, all of them aimed against intracellular protein structures. GAD65 as molecule with intracellular localization as well as the neuropathological admixture of neurodegeneration with invasion of T-lymphocytes in the biopsy tissue of patients (Malter et al., 2010) may suggest analogies of the present disease pattern with paraneoplastic neurological syndromes. Here, specific onconeuronal antigens including Ta/M proteins are expressed constitutively in neurons and ectopically in the associated tumors. In brain lesions, neurodegeneration is accompanied by perivascular infiltrates constituted of CD4+ T-cells and B-cells, whereas CD8+ T-cells predominate within the CNS parenchyma (Voltz et al., 1998; Rosenfeld et al., 2001). Purely onco-neuronal antibody-dependent mechanisms are doubtful in view of the intracellular localization of most onconeuronal antigens and a consistent failure to replicate the CNS pathology of paraneoplastic neurological syndromes by antibody transfer in an animal model (Sillevis Smitt et al., 1995; Tanaka et al., 1995; Carpentier et al., 1998). More recently, pathological and immunological investigations implicate the T-cell component of the lesions as the primary pathogenic principle in these disorders (Albert et al., 1998; Voltz et al., 1998; Benyahia et al., 1999; Plonquet et al., 2002). In an exemplary fashion, it was shown that by adoptive transfer T-cells specific for the autologous onco-neuronal antigen Pnma1 elicited a strong neuroimmunological reaction in contrast to minor pathology by anti-Pnma1 antibodies induced by protein immunization. These data indicate that in paraneoplastic neurological syndromes pathogenetically critical T-lymphocytes can target identical molecules as the corresponding onconeuronal antibody (Pellkofer et al., 2004). Significant impairing effects of anti-GAD65 antibodies to neurons and transmission have not been found in experimental conditions (Stemmler et al., 2015; Hackert et al., 2016). These considerations are now included in the revised manuscript (p. 20, ll. 451-476).

References

Albert ML, Darnell JC, Bender A, Francisco LM, Bhardwaj N, Darnell RB Tumor-specific killer cells in paraneoplastic cerebellar degeneration. Nat Med 1998; 4:1321-1324.

Benyahia B, Liblau R, Merle-Beral H, Tourani JM, Dalmau J, Delattre JY Cell-mediated autoimmunity in paraneoplastic neurological syndromes with anti-Hu antibodies. Ann Neurol 1999; 45:162-167.

Blumcke I et al. Histopathological Findings in Brain Tissue Obtained during Epilepsy Surgery. N Engl J Med 2017; 377:1648-1656.

Carpentier AF, Rosenfeld MR, Delattre JY, Whalen RG, Posner JB, Dalmau J DNA vaccination with HuD inhibits growth of a neuroblastoma in mice. Clin Cancer Res 1998; 4:2819-2824.

Graus F et al. A clinical approach to diagnosis of autoimmune encephalitis. Lancet Neurol 2016; 15:391-404.

Gresa-Arribas N, Titulaer MJ, Torrents A, Aguilar E, McCracken L, Leypoldt F, Gleichman AJ, Balice-Gordon R, Rosenfeld MR, Lynch D, Graus F, Dalmau J Antibody titres at diagnosis and during follow-up of anti-NMDA receptor encephalitis: a retrospective study. Lancet Neurol 2014; 13:167-177.

Hackert JK, Muller L, Rohde M, Bien CG, Kohling R, Kirschstein T Anti-GAD65 Containing Cerebrospinal Fluid Does not Alter GABAergic Transmission. Front Cell Neurosci 2016; 10:130.

Malter MP, Helmstaedter C, Urbach H, Vincent A, Bien CG Antibodies to glutamic acid decarboxylase define a form of limbic encephalitis. Ann Neurol 2010; 67:470-478.

McDonald JH (2014) Handbook of biological statistics Sparky House Publishing.

Pellkofer H, Schubart AS, Hoftberger R, Schutze N, Pagany M, Schuller M, Lassmann H, Hohlfeld R, Voltz R, Linington C Modelling paraneoplastic CNS disease: T-cells specific for the onconeuronal antigen PNMA1 mediate autoimmune encephalomyelitis in the rat. Brain 2004; 127:1822-1830.

Pitsch J, Kamalizade D, Braun A, Kuehn JC, Gulakova PE, Ruber T, Lubec G, Dietrich D, von Wrede R, Helmstaedter C, Surges R, Elger CE, Hattingen E, Vatter H, Schoch S, Becker AJ Drebrin Autoantibodies in Patients with Seizures and Suspected Encephalitis. Ann Neurol 2020.

Plonquet A, Gherardi RK, Creange A, Antoine JC, Benyahia B, Grisold W, Drlicek M, Dreyfus P, Honnorat J, Khouatra C, Rouard H, Authier FJ, Farcet JP, Delattre JY, Delfau-Larue MH Oligoclonal T-cells in blood and target tissues of patients with anti-Hu syndrome. J Neuroimmunol 2002; 122:100-105.

Rosenfeld MR, Eichen JG, Wade DF, Posner JB, Dalmau J Molecular and clinical diversity in paraneoplastic immunity to Ma proteins. Ann Neurol 2001; 50:339-348.

Sillevis Smitt PA, Manley GT, Posner JB Immunization with the paraneoplastic encephalomyelitis antigen HuD does not cause neurologic disease in mice. Neurology 1995; 45:1873-1878.

Stemmler N, Rohleder K, Malter MP, Widman G, Elger CE, Beck H, Surges R Serum from a Patient with GAD65 Antibody-Associated Limbic Encephalitis Did Not Alter GABAergic Neurotransmission in Cultured Hippocampal Networks. Front Neurol 2015; 6:189.

Tanaka M, Tanaka K, Onodera O, Tsuji S Trial to establish an animal model of paraneoplastic cerebellar degeneration with anti-Yo antibody. 1. Mouse strains bearing different MHC molecules produce antibodies on immunization with recombinant Yo protein, but do not cause Purkinje cell loss. Clin Neurol Neurosurg 1995; 97:95-100.

Voltz R, Dalmau J, Posner JB, Rosenfeld MR T-cell receptor analysis in anti-Hu associated paraneoplastic encephalomyelitis. Neurology 1998; 51:1146-1150.

Wright SP Adjusted P-Values for Simultaneous Inference. Biometrics 1992; 48:1005-1013.

---

## [Decision Letter · Decision Letter 1]

13 Oct 2020

Adult-onset temporal lobe epilepsy suspicious for autoimmune pathogenesis: autoantibody prevalence and clinical correlates

PONE-D-20-21286R1

Dear Dr. Pitsch,

We’re pleased to inform you that your manuscript has been judged scientifically suitable for publication and will be formally accepted for publication once it meets all outstanding technical requirements.

Kind regards,

Giuseppe Biagini, MD

Academic Editor

PLOS ONE

Additional Editor Comments (optional):

Reviewers' comments:

Reviewer's Responses to Questions

**Comments to the Author**

1. If the authors have adequately addressed your comments raised in a previous round of review and you feel that this manuscript is now acceptable for publication, you may indicate that here to bypass the “Comments to the Author” section, enter your conflict of interest statement in the “Confidential to Editor” section, and submit your "Accept" recommendation.

Reviewer #1: All comments have been addressed

2. Is the manuscript technically sound, and do the data support the conclusions?

Reviewer #1: Yes

3. Has the statistical analysis been performed appropriately and rigorously? 

Reviewer #1: Yes

4. Have the authors made all data underlying the findings in their manuscript fully available?

Reviewer #1: Yes

5. Is the manuscript presented in an intelligible fashion and written in standard English?

Reviewer #1: Yes

6. Review Comments to the Author

Reviewer #1: All of my comments have been adressed. I have no further comments. This is a very interesting study on a very large material, and as such it is a valuable addition to the literature.

7. PLOS authors have the option to publish the peer review history of their article (what does this mean?). If published, this will include your full peer review and any attached files.

Reviewer #1: No

---

## [Editor Report · Acceptance letter]

19 Oct 2020

PONE-D-20-21286R1 

Adult-onset temporal lobe epilepsy suspicious for autoimmune pathogenesis: autoantibody prevalence and clinical correlates 

Dear Dr. Pitsch:

I'm pleased to inform you that your manuscript has been deemed suitable for publication in PLOS ONE. Congratulations! Your manuscript is now with our production department. 

Kind regards, 

on behalf of

Dr. Giuseppe Biagini 

Academic Editor

PLOS ONE